# Are Basic Substances a Key to Sustainable Pest and Disease Management in Agriculture? An Open Field Perspective

**DOI:** 10.3390/plants12173152

**Published:** 2023-09-01

**Authors:** Silvia Laura Toffolatti, Yann Davillerd, Ilaria D’Isita, Chiara Facchinelli, Giacinto Salvatore Germinara, Antonio Ippolito, Youssef Khamis, Jolanta Kowalska, Giuliana Maddalena, Patrice Marchand, Demetrio Marcianò, Kata Mihály, Annamaria Mincuzzi, Nicola Mori, Simone Piancatelli, Erzsébet Sándor, Gianfranco Romanazzi

**Affiliations:** 1Dipartimento di Scienze Agrarie e Ambientali (DiSAA), Università degli Studi di Milano, Via Celoria 2, 20133 Milano, Italy; giuliana.maddalena@unimi.it (G.M.); demetrio.marciano@unimi.it (D.M.); 2Institut de l’Agriculture et de l’Alimentation Biologiques (ITAB), 149 rue de BERCY, F-75012 Paris, France; yann.davillerd@itab.asso.fr (Y.D.); patrice.marchand@itab.asso.fr (P.M.); 3Dipartimento di Scienze Agrarie, Alimenti, Risorse Naturali e Ingegneria (DAFNE), University of Foggia, Via Napoli 25, 71122 Foggia, Italy; ilaria.disita@unifg.it (I.D.); giacinto.germinara@unifg.it (G.S.G.); 4Department of Biotechnology, University of Verona, Strada le Grazie 15, 37134 Verona, Italy; chiara.facchinelli@univr.it (C.F.); annamaria.mincuzzi@uniba.it (A.M.); nicola.mori@univr.it (N.M.); 5Department of Soil, Plant and Food Sciences, University of Bari Aldo Moro, Via Amendola 165/A, 70126 Bari, Italy; antonio.ippolito@uniba.it; 6Agricultural Research Center, Plant Pathology Research Institute, 9 Gamaa St., Giza 12619, Egypt; youssefeladawy@yahoo.com; 7Department of Organic Agriculture and Environmental Protection, Institute of Plant Protection–National Research Institute, Władysława Wêgorka 20, 60-318 Poznañ, Poland; j.kowalska@iorpib.poznan.pl; 8Faculty of Agricultural and Food Science and Environmental Management, Institute of Food Science, University of Debrecen, Böszörményi út 138, 4032 Debrecen, Hungary; mihaly.kata@agr.unideb.hu (K.M.); karaffa@agr.unideb.hu (E.S.); 9Department of Agricultural, Food and Environmental Sciences, Marche Polytechnic University, Via Brecce Bianche 10, 60131 Ancona, Italy; s.piancatelli@pm.univpm.it (S.P.); g.romanazzi@univpm.it (G.R.)

**Keywords:** disease management, pest management, sustainable crop protection, integrated pest management, organic farming

## Abstract

Pathogens and pests constantly challenge food security and safety worldwide. The use of plant protection products to manage them raises concerns related to human health, the environment, and economic costs. Basic substances are active, non-toxic compounds that are not predominantly used as plant protection products but hold potential in crop protection. Basic substances’ attention is rising due to their safety and cost-effectiveness. However, data on their protection levels in crop protection strategies are lacking. In this review, we critically analyzed the literature concerning the field application of known and potential basic substances for managing diseases and pests, investigating their efficacy and potential integration into plant protection programs. Case studies related to grapevine, potato, and fruit protection from pre- and post-harvest diseases and pests were considered. In specific cases, basic substances and chitosan in particular, could complement or even substitute plant protection products, either chemicals or biologicals, but their efficacy varied greatly according to various factors, including the origin of the substance, the crop, the pathogen or pest, and the timing and method of application. Therefore, a careful evaluation of the field application is needed to promote the successful use of basic substances in sustainable pest management strategies in specific contexts.

## 1. Introduction

The use of plant protection products, such as fungicides, insecticides, and herbicides, is crucial for controlling diseases and pests in agriculture, but their safety, costs, and availability are a growing concern [1,2,3,4]. The possible adverse effects on human and environmental health have led to the development of risk exposure indicators [5,6] and more stringent legislative requirements [7,8]. The EU, for example, regulates plant protection products authorization [9] and utilization to endorse a new paradigm for agricultural production with the transition to low-input farming, promoting integrated pest management and complementary alternatives to minimize the utilization of plant protection products [10]. The use of plant protection products has negative impacts, in terms of their direct costs and negative externalities, on producers and the environment, especially in developing countries [11]. While ensuring rigorous testing for safety and quality, the product registration process increases the costs of developing new products and lengthens the time to market [12,13]. Additionally, the shift towards single-site compounds, which have a more favorable profile than multi-site compounds, increases the risk of resistance development in pests and pathogens [14].

Basic substances can represent an opportunity to mitigate the problems associated with traditional plant protection products. Basic substances are defined as compounds that are not predominantly used as plant protection products but may be useful in crop protection. They have no toxicological concerns and do not cause adverse effects on humans, animals, or the environment [9]. Interestingly, ‘foodstuff’ substances (as defined by Regulation (EC) No. 178/2002) are intrinsically considered basic substances [15]. Basic substances have no residue limits, and usually no pre-harvest interval [16]. Also, since they are not currently placed on the market as plant protection products, they are not considered in the Harmonized Risk Indicator 1 calculation that is used in the EU for highlighting the trends in the risks associated with the use of pesticides [17]. European basic substances partially overlap with the American “Generally Recognized as Safe” (GRAS) substances, which are approved for use in food products as preservatives [18].

To date, the European pesticide database (https://ec.europa.eu/food/plant/pesticides/eu-pesticides-database/start/screen/active-substances, accessed on 28 March 2023) lists 24 approved active substances, and reports on crop and pest indications, formulation, type, rate, and phenological stage of application. These approved basic substances have a wide range of applications. They can act as direct control products for diseases and pests by exerting a fungicidal, bactericidal, or insecticidal activity, or they can be employed in indirect control strategies, such as in triggering the plant immune response (e.g., elicitors), or as attractants or repellents (Appendix A). However, no indication has been provided on the expected field efficacy or their integration in disease and pest management programs. According to the FAO [19], sustainable food and agriculture should contribute to the three dimensions of sustainability: environmental, social, and economic. In this context, the management of crop pests and diseases should not only consider the costs of protection but also the efficacy of protection, which influences the yield and, consequently, the economic dimension of sustainability. Therefore, a careful evaluation of the outcomes of basic substances’ employment in the open field is highly necessary. Fortunately, the literature regarding both the approved and potential basic substances is continuously growing, as evidenced from scientific studies and review papers [20,21,22,23,24,25,26,27] (Figure 1).

With this review, we aimed to provide the reader with updated information concerning the use of approved and potential basic substances for crop protection under field conditions, to summarize the findings achieved in the recent past, and provide indications on the exploitation and integration of basic substances in effective disease management programs. The information available on the practical aspects and field applications arising from the basic substances literature was integrated with that available in the EU pesticides database, taking into consideration specific case studies on the use of basic substances for controlling diseases and pests.

## 2. Activity of Approved Basic Substances against Fungal Diseases

Fungicides have been the top-selling group of plant protection products in the EU for a long time. Three countries, namely Spain, France, and Italy, make up around 62% of the total volume of pesticides (330 thousand tons) sold annually between the years 2011 and 2020 [28]. Interestingly, these countries also have the highest surface area dedicated to viticulture, which accounts for 75% of the 3.2 million hectares under vines [29]. This is due to the fact that many fungicide sprays are applied each growing season to the grapevine crop (*Vitis vinifera*) to manage three major fungal diseases: downy and powdery mildews, and grey mold [30]. The major fungicide markets are fruit and vegetables, cereals, grapevines, and potatoes, which account for about 60% of the global fungicide market (https://www.apsnet.org/edcenter/apsnetfeatures/Pages/Fungicides.aspx, accessed on 3 May 2023). In the following paragraphs, information on the results achieved through basic substance applications to control the important diseases of grapevines, potatoes, and fruits, in general, will be provided, taking into consideration that pre-harvest treatments also affect the post-harvest control of the pathogens (Table 1).

### 2.1. Grapevine

#### 2.1.1. Grapevine Downy Mildew

European grapevine exhibits a high level of susceptibility to grapevine downy mildew, caused by the oomycete *Plasmopara viticola* [31]. To prevent infections and the consequent production loss, several treatments with chemical fungicides are needed during the season under both organic and integrated pest management systems [31]. This results in negative consequences for the environment and risks for human health. Copper is the most widely applied plant protection product acting against grapevine downy mildew, although the Regulation (EU) 2018/1981 restricted the quantities allowed and classified this heavy metal as an active substance candidate for substitution [32]. Copper fungicides are fundamental for organic productions, where the use of synthetic curative compounds is not allowed, but also play a central role in integrated pest management to limit the outbreak of resistant strains. This situation encouraged the search for alternative tools to protect plants from *P*. *viticola*. Chitosan [33], biocontrol agents [34], aptamers [35], hydrolyzed proteins [36], laminarin [37], stilbenes [38], and other plant extracts [39,40,41] showed promising results under in vitro or in vivo experiments. Among these alternatives, basic substances could present a good opportunity. Chitosan, *Equisetum arvense* (horsetail), sucrose, *Salix* spp. cortex, lecithin, fructose, and nettle *(Urtica* spp.) are the basic substances that may exhibit effectiveness against grapevine downy mildew [16], especially when integrated into reduced copper strategies. The field application of chitosan hydrochloride alone showed promising results in plot trials, under different environmental conditions, and even under the presence of a high disease pressure [22,42,43]. This biopolymer is obtained from chitin deacetylation, and it can perform eliciting, antimicrobial, and film-forming activities once applied on plant tissues [22]. Results obtained with chitosan individual treatments were similar to those obtained with a conventional application of copper, showing disease reductions compared to the untreated control, which in some cases exceeded 95% on leaves and 80% on grape bunches [22,42]. Nevertheless, chitosan effectiveness against grapevine downy mildew is strictly linked to two main factors: volume of applications and active ingredient concentrations. To best perform the triple mode of action on plant tissues, a good wetting of the canopy is required and the standard spraying volume for grapevine (1000 L/ha of water) is recommended. Application of 0.8% chitosan has been found to perform better than copper hydroxide in seasons characterized by frequent rainfall and high disease pressure [42]. The 0.5% of active ingredient does not usually show significant differences compared to the 0.8% in terms of their efficacy, as well as being less expensive for the growers. Furthermore, treatments with high concentrations of chitosan for the whole season can induce undesired collateral physiological responses in vines, such as reduced growth and leaf area [42]. In addition to being dangerous for humans and ecosystems [44], copper residues on the berries affect the wine quality, reducing the concentration of several amino acids in the must [45]. Unlike copper, chitosan and other natural compounds, such as laminarin, have lower impacts on the final product quality [45]. Results obtained in the past years have suggested chitosan as a promising tool to support or eventually replace copper for grapevine downy mildew management. Copper and chitosan could even coexist to begin with, for example with alternating or combined treatments, even if validations on a commercial scale for these strategies are needed. According to the data available in the literature, no copper could be applied under instances of low disease pressure, while a valid strategy for difficult seasons could be to apply copper until flowering (in the period of higher susceptibility) and then replace it with chitosan. In this way, it could be possible to reduce the quantities of copper distributed per year on the one hand and the costs of chitosan on the other hand. Indeed, the main limitations regarding chitosan diffusion so far are represented by its cost and the lack of operational knowledge. It will be important to invest in new formulations and to investigate the miscibility of this biopolymer with other plant protection products, since farmers are used to simultaneously applying several compounds so as to target different pests within a single treatment.

#### 2.1.2. Grey Mold on Table Grape

Grey mold is a globally widespread and economically relevant disease of grapes caused by the second most important phytopathogenic fungus, *Botrytis cinerea* [46]. This broad host range pathogen affects several crops, both under pre- and post-harvest. *B. cinerea* can survive and develop in vineyards as both a necrotrophic pathogen and a saprophyte [47,48]. Grey mold can result from multiple infection pathways on ripening grape berries, including latent infections established during blooming, direct berry infection due to airborne conidia, and berry-to-berry infection caused by mycelium originating from previously infected berries (nesting path) within the cluster [49,50], which spread according to a nesting path. Although *B. cinerea* causes about 30% of latent infections [51], it is difficult to precisely estimate the global losses due to its broad host range and specific missing statistics. New Zealand recorded costs due to grey mold direct crop losses and grey mold control measures of up to NZD 5000/ha and NZD 1500/ha, respectively, in growing seasons favorable for disease development [47]. In Australia, Chile, and South Africa, grey mold is the main cause of wine and table grape losses, from the vineyard to the retail outlet, entailing profit reductions of AUD 52 million/year, USD 22.4 million/year, and ZAR 25 million/year, respectively [46]. Chemical fungicides are the most important control means available, although fungicide resistance is an increasing issue for this pathogen [52,53,54]. Fungicide-resistant phenotypes were detected in *B. cinerea* populations in table grape vineyards in California, with genotypic resistance against boscalid, cyprodinil, fenhexamid, and pyraclostrobin in 95%, 85%, 23%, and 14% of tested isolates, respectively [55]. Differences in the fungicide resistance profile of *B. cinerea* may be due to the species/groups included within the complex; as an example of biodiversity, in the pomegranate fruit, *B. cinerea*, *B. pseudocinerea*, and *Botrytis* group *S* were the etiological agents of grey mold [56,57]. Currently, latent infections caused by *Botrytis* spp. are completely prevented in storage through the use of SO_2_-generating pads [51], although these entail adverse effects on food, humans, and the environment (i.e., phytotoxicity, development of antimicrobial resistance, allergy, pollution, etc.) [58], and cannot be applied to organic table grapes. This encourages the set-up of new, safer, more effective, and cheaper alternative control means and strategies. Basic substances, such as salts and chitosan, and potential basic substances can be a promising alternative to chemical fungicides for grey mold management [59,60]. Chitosan treatments have been shown to significantly reduce disease incidence both in the field and after harvest. It indirectly enhances the activity of the key plant enzymes involved in disease resistance, such as superoxide dismutase, peroxidase, catalase, and ascorbate peroxidase, that damage the mycelial structures of *Botrytis* spp. and reduce pathogen development [61,62,63].

Grey mold on table grapes is a disease affecting clusters both in the field and during the post-harvest phases. Unfortunately, since *B. cinerea* affects grapes more heavily during the post-harvest phases, most of the papers that are available on this subject concern the disease development after harvest, and very few concern pre-harvest evaluations [64]. Grey mold protection starts during the grapevine growing season following a well-established scheme, in which four applications of fungicides are carried out under the following specific phenological stages: berry set, pre-bunch closure, veraison, and 1–3 weeks before harvest. This strategy, that is mandatory to avoid latent infections during the growing season [50], was also adopted for chitosan and other alternative control means. In field treatments on table grapes, 1% chitosan demonstrated the same ability to protect grapes from grey mold as the strategy based on synthetic fungicide application. In an integrated program lasting two years, chitosan-treated “Chardonnay” wine grapes exhibited a degree of disease severity at harvest that was more than halved compared to the untreated control and was as effective as the synthetic fungicide program [65]. Chitosan has also been combined with active antimicrobial substances, such as essential oils, and applied as pre-harvest treatments [66] or as post-harvest coatings to improve the preservation of qualitative parameters and reduce the product losses caused by *Botrytis* spp. A possible evolution in the application of chitosan is its formulation as nanoparticles, which in preliminary trials behaved better than standard formulations [63]. This basic substance has been used formulated as a chitosan/silica nanocomposite-based compound, which reduced conidial germination and germ tube elongation, affecting the development of grey mold on the grapes [63]. Various substances have been tested at pre-harvest in combination with chitosan, such as with chitosan added or complexed with salicylic acid. In particular, the CTS-g-SA complex improved fruit physiology (transpiration and respiration rates), qualitative parameters (soluble solids, titratable acidity, and total phenolic content), and defense mechanisms involving the control of disease incidence [58].

Among the other basic substances currently approved, sodium bicarbonate has been studied worldwide, leading to results that are, in many cases, not different, if not better, than the synthetic fungicides [67]. When applied before harvest, sodium bicarbonate showed a significant reduction in botrytis storage rots in both small-scale and large-scale tests. In large-scale trials simulating practical commercial conditions adopted in Southern Italy, two salt applications (at 30 and 90 days before harvest) of sodium bicarbonate significantly reduced grey mold from 23% (untreated control) to 12% [68]. Among these basic substances, sodium bicarbonate may represent one of the most useful and effective compounds, considering that it is easy to find on the market, is very cheap, has a broad spectrum of activity against a variety of pathogens, is well accepted by consumers and operators, and has an acceptable environmental profile.

### 2.2. Potato Leaf Diseases

The potato (*Solanum tuberosum*) is one of the most important vegetable crops in the world. It belongs to the family Solanaceae and is an important starchy food crop. Potato plants are subjected to numerous diseases wherever the crop is grown. Among the approved basic substances, there are some that have the potential to limit the early and late blight of potatoes by spraying or dipping tubers before sowing. Early blight of potatoes caused by *Alternaria solani* and *Alternaria alternata*, and late blight caused by the oomycete *Phytophthora infestans* are major causes of concern in potato production. This problem is particularly important in organic farming, where synthetic fungicides are prohibited. Therefore, a necessary condition in the organic cultivation of potatoes is the timely implementation of treatments and adherence to the rules of agricultural technology regarding the appropriate variety and crop rotation.

*Alternaria* spp. are air- and soil-borne organisms that cause disease on foliage (leaf blight), stems (collar rot), and tubers (tuber rot), resulting in severe damage during all stages of plant development [69]. This disease causes losses in crop productivity in the field and in tuber quality during storage. The average annual yield loss of potatoes due to this disease is approximately 79% of the total production, depending on the nature of the disease, weather conditions, and the type of variety grown [70]. It can destroy foliage prematurely and in a short time, reducing production, while the tuber infections associated with rots can cause significant crop losses during storage. It is considered one of the most destructive crop pathogens threatening global food security. In organic potato production, late blight can cause severe losses in the potato yield and quality. Currently, in organic farming it can only be effectively controlled using copper fungicides. However, some countries prohibit copper use in organic farming based on their national laws, as the harmfulness of copper in ecosystems is still being debated. Studies aiming at the reduction in copper usage and testing of potential basic substances against late blight for organic farming are needed. Currently, basic substances, such as extracts from the bulb of the onion crop (*Allium cepa*) and horsetail, have been proposed as protective treatments against early blight. In the case of *P. infestans,* chitosan hydrochloride is most frequently mentioned as an elicitor, along with nettle extracts, lecithins, and dried horsetail [16]. However, the use of these substances in the field has not often been demonstrated. The application of elicitors and botanical fungicides, beneficial microorganisms, and basic substances should be a combination of compounds and microorganisms with different modes of action, beginning at the early stages of the potato plant’s growth [71,72,73,74]. This strategy of minimizing the risk to the ecosystem is a global trend, especially in the EU, where the policy of greening agriculture is being promoted.

Chitosan significantly inhibits the mycelial growth and in vitro spore germination of *P. infestans*, induces resistance to the pathogen in potato pieces and leaves [75], and forms a mechanical barrier to the pathogen penetration [76,77]. It also has a synergistic effect with plant protection products, making it a potential way to reduce the use of chemical plant protection products. In field conditions, the use of chitosan can stimulate plants to defend themselves, which in turn contributes to limiting the harmful effects of potato disease symptoms. Late blight epidemics were delayed on plots that received eight sprays of 0.1% chitosan [78] and provided 60% protection against late blight by mixing 4% chitosan with a plant elicitor [79]. Some late blight reducing potential for 0.4% chitosan was found in field tests performed in Germany on the cultivars Nicola and Ditta [80]. Field tests confirmed some of the major results coming from the lab and growth chamber assays. Most effects were only visible during the early phases of the disease, when plants were still vigorous, but might have been more pronounced under a different infection regime with an earlier onset of the disease. Both in full-field tests, chitosan (0.4%) and the copper fungicide, and in a small plot trial, chitosan (0.4%) accompanied with the horsetail and liquorice products seemed to be able to cause some degree of disease reduction, even under an extremely late infection regime [80]. Good results with a low-level copper formulation (copper sulfate pentahydrate), together with chitosan as an adhesive substance to increase rain fastness, were also obtained [81]. In recent years, practical applications of chitosan were tested against *P. infestans* in in vivo experiments under outdoor conditions [82]. This experiment showed that chitosan is very effective against *P. infestans*. An average damage of over 76% was observed in the control plants. In the treated variants with 1–4 applications of chitosan, the final damage to the plants ranged from 48% to 0.5%. Expressed as values of the final inhibitory protective effect, a single application of a 0.4% solution of chitosan provided an inhibitory effect of 37%. In cases where chitosan was applied four times, an inhibitory effect of up to 99.3% was demonstrated [82]. The newest study also confirmed that chitosan can be applied as the nano compound. The bioactivity and absorbency of elicitors are critical factors that limit the large-scale field application. A star polymer was constructed to deliver the nano-sized (particle size from 144.61 nm to 17.40 nm in an aqueous solution) chitosan to enhance the control effects against potato late blight [83].

As basic substances, lecithins have fungicidal activity due to their inhibition of the fungal hypha penetration into the plant cells. Unlike chitosan, lecithins are not fully soluble in water. Since chitosan would be the first choice in most situations when looking for a fungicide among the basic substances, the next option for controlling oomycetes may be to use lecithins in combination with chitosan in joint field treatments (https://eutrema.co.uk/basic-substances-what-are-they-and-how-can-they-used-for-pest-and-disease-control-on-farms/, accessed on 20 January 2023). The fatty acids present in the lecithins could act more positively via plant defense stimulation, rather than through a toxic effect. In fact, linolenic acid and its precursor linoleic acid, both present in soy lecithin, are the precursors of a wide variety of oxylipins and the plant hormone jasmonic acid, which actively participate in plant defenses [84]. Lecithins have not been studied in field trials.

In field tests, the application of 12 kg/hL horsetail macerate showed effectiveness in protecting the tomato crop (*Solanum lycopersicum*) from late blight that was analogous to the copper-based treatments [20].

Nettle slurry (*Urtica dioica*), used as a foliar fertilizer in different doses, alone or in combination with horsetail, had no significant effects on the yield, chlorophyll content, or the presence of pests and diseases in organic potato crops [85]. Conversely, the methanolic leaf extracts of nettle slurry and broad-leaf hopbush (*Dodonaea viscosa*) demonstrated a strong antifungal efficacy against *A. alternata*. Among the many polyphenolic compounds that were detected in the HPLC of the extract, coumaric acid, caffeic acid, ferulic acid, and α-tocopherol showed potent in vitro fungicidal activity against *A. alternata*, either applied alone or in combination at low concentrations [86].

In Romania, 2.2% and 3.3% water solutions of the onion crop showed significant protection against *A. solani* in potato fields [87]. Dry extracts of onion (concentration 20.0 mg/mL) showed antifungal activity against *A. alternata* and *P. infestans.* In particular, red onion extracts showed a higher efficacy in inhibiting *A. alternata* than white onion extracts, which showed no efficacy. This result is surprising, considering that both extracts have a similar amount of quercetin, an antioxidant with antifungal activity. Evidently, other components of these extracts are responsible for the *A. alternata* inhibition [88].

### 2.3. Pre-Harvest Treatment Affecting the Post-Harvest Diseases of Fruits

Fruit-bearing plants may be infected in the field before or during their harvest, providing inoculum for post-harvest decay following their harvest [89]. The accumulation and/or survival of the inoculum can be prevented through pre-harvest treatments [90]. Basic substances may provide environmentally friendly alternatives to pre-harvest fungicide application to prevent the post-harvest decay of fruits and vegetables, although there is limited information about their efficacy.

Chitosan hydrochloride and chitosan are the most widely studied basic substances in pre-harvest application, either alone or in combination. Similarly to what has been observed for table grapes [91,92,93,94], the pre-harvest application of 0.2–1% chitosan was effective against grey mold latent infection and the decay of strawberries (*Fragaria* x *ananassa* and *Fragaria chiloensis*) [95,96,97,98,99]. The pre-harvest application of 1% chitosan was effective against the grey mold and brown rot of sweet cherries (*Prunus avium*) and date palm fruits (*Phoenix dactylifera*) [100,101]. The soft rot of kiwifruit (*Actinidia deliciosa*) caused by *Botryosphaeria dothidea* and *Phomopsis* sp. was also reduced following a chitosan-containing spray [102]. This basic substance was effective in being able to reduce the *A. alternata*-related decay of apricots (*Prunus armeniaca*) [103,104] and in the decay of peaches (*Prunus persica*) [105,106]. The pre-harvest treatment of jujube (*Zizyphus jujuba*) and tomato plantations with 0.3–1 g/L chitosan was also effective in being able to reduce the decay of these harvested fruits [107,108]. However, in the case of raspberries (*Rubus idaeus*), 1% or 2% chitosan was only effective in reducing the decay of these fruits during their storage [109].

**Table 1 plants-12-03152-t001:** List of the basic substances which effectively protected the crops described in the present study from specific diseases.

Crop (Species)	Disease (Pathogen)	Basic Substance	Reference
Grapevine (*Vitis vinifera*)	Downy mildew (*Plasmopara viticola*)	Chitosan	[42,43]
	Botrytis bunch rot (*Botrytis cinerea*)	Chitosan	[66]
Potato (*Solanum tuberosum*)	Early blight (*Alternaria alternata*)	Nettle slurry (*Urtica dioica*) and broad-leaf hopbush (*Dodonaea viscosa*) methanolic extracts	[86]
	Early blight (*Alternaria solani*)	Water solutions of *Allium cepa*	[87]
	Late blight (*Phytophthora infestans*)	Chitosan	[78,79,80,82]
Strawberry (*Fragaria* x *ananassa* and *Fragaria chiloensis*)	Grey mold (*Botrytis cinerea*)	Chitosan	[90,95,96,97]
Sweet cherry (*Prunus avium*)	Storage decay	Chitosan	[98]
	Botrytis rot (*B. cinerea*)	Sodium bicarbonate salts	[108]
Date palm fruit (*Phoenix dactylifera*)	Storage decay	Chitosan	[99]
Kiwifruit (*Actinidia deliciosa*)	Soft rot (*Botryosphaeria dothidea* and *Phomopsis* sp.)	Chitosan	[100]
Apricot (Prunus armeniaca)	Decay (*A. alternata*)	Chitosan	[101,102]
Peach (Prunus persica)	Decay (*A. alternata*)	Chitosan	[103,104]
Jujube (*Zizyphus jujuba*)	Storage decay	Chitosan	[105]
Tomato (*Solanum lycopersicum*)	Storage decay	Chitosan	[106]
Pear (*Pyrus communis*)	Storage decay	Onion (*Allium cepa*) extract	[109]

The pre-harvest application of other basic substances has received fewer widespread studies. The botrytis rot on sweet cherries was reduced by spraying sodium bicarbonate salts, even if with a far lower effectivity than under post-harvest application [110]. Three applications of onion extract on pear trees (*Pyrus communis*) decreased the decay of the stored fruits [111].

The pre-harvest effectivity of basic substances with effective control of post-harvest pathogens, such as horsetail extract that protects from post-harvest pathogens, like *B. cinerea, C. acutatum*, and *Monilinia* sp. [16], should also be studied in the future.

The pre-harvest usage of basic substances provides a promising alternative to fungicides acting against different post-harvest pathogens. However, their application must be optimized for different plant products and conditions. Their effect, for instance, can be further increased in combination with different substances, like calcium, salicylic acid, or methyl jasmonate [100,101,103,104,105,112,113,114].

## 3. Activity of Approved Basic Substances against Insects

Based on the analysis of data obtained from Scopus, over 3000 articles regarding eco-friendly natural product pesticides in crop protection have been published in the Agricultural and Biological Sciences sector, with an increasing trend in the last 30 years and a peak of 271 papers published in 2021. However, a search on Scopus using the keywords ‘pest control’ and ‘basic substance’ yielded only 11 scientific articles published since 2015, with the first article published in 2015 [15]. According to the EU Regulation (EC) 1107/2009, among the twenty-four basic substances permitted for plant protection use, eight are approved as insecticides (nettle, sodium chloride, L-cysteine, sucrose, and fructose), physical barriers (talc E553B), attractants (diammonium phosphate), or repellents (onion oil). In the following paragraphs, the literature concerning the field application of these eight basic substances will be described, along with their modes of action (Figure 2).

### 3.1. Nettle

The extracts of nettle, commonly known as a foodstuff and medicine, are traditionally used by farmers who claim a significant reduction in the aphid and Coleoptera presence [115,116,117]. Searching the keywords “*Urtica*” and “pest management” on the Scopus database resulted in the retrieval of more than 500 papers that were published in the Agricultural and Biological Sciences sector, demonstrating that nettle is one of the most studied basic substances for pest management purposes. Nettle can be used as a fermented aqueous extract in spray applications against different aphid species, such as *Myzus persicae*, *Macrosiphum rosae*, *Eriosoma lanigerum*, and *Panaphis juglandi*, to protect fruit trees (*Malus domestica*, *Prunus* spp.), elder trees, beans (e.g., *Phaseolus vulgaris*), leafy vegetables (*Lactuca sativa*, *Brassica oleracea*), *Rosa* spp., and *Spiraea* spp. With a population density reduction of more than 30%, nettle extracts can also be used on Brassicaceae crops against the flea beetle, *Phyllotreta nemorum*, and the diamond back moth, *Plutella xylostella*, as well as on apple and pear trees against the codling moth, *Cydia pomonella*. In field trials, nettle slurry fermented extract showed a repellent activity towards *Hyalopterus pruni* and *P. juglandi* [118,119], but not against *Aphis spiraephaga* [118], suggesting that the efficacy of the nettle slurry extract against aphids is species-dependent. Under controlled conditions, *Urtica urens* water extract effectively limited the fertility of *M. persicae,* slightly reducing the increase of its population (by 20% on average), while no negative effects were registered on its natural enemy, *Macrolophus pygmaeus* [115]. Furthermore, the nettle extract used in combination with other biorational insecticides could improve the efficacy against aphid pests [115]. Nettle extracts can also be used to control the mites *Tetranychus urticae* on beans and *Tetranychus telarius* on grapevines. Repellent, acaricidal, and antifeedant activities of the nettle extracts against *T. urticae*, one of the economically most important pests in a wide range of outdoor and protected crops worldwide, have been reported [120,121]. To the best of our knowledge, less information is available in the literature regarding the effect of nettle extracts on *T. telarius*. In this scenario, extracts from several plants proved to exert insecticidal or miticidal activity against vegetables and stored-product pests [122,123,124,125], that, in some cases, were comparable to those achieved using chemical insecticides (e.g., synthetic pyrethroid) [126], and could suggest potential basic substances as alternatives to synthetic chemical insecticides in crop protection.

### 3.2. Sucrose and Fructose

Sucrose and fructose are involved in the phenomenon of “Sweet Immunity”, according to which the sugar metabolism and signaling influence the plant immunity networks [21,127,128]. The quantities and ratios of three soluble carbohydrates (sucrose, D-fructose, and glucose) and three sugar alcohols (sorbitol, quebrachitol, and myo-inositol) of apple tree surfaces play a role in the trees’ resistance, as they influence the host preference, egg laying, and the behavior of the neonate larvae of *Cydia pomonella* [129,130,131,132]. Recent studies demonstrated that sucrose, in micro-dose foliar applications, can induce partial resistance via antixenosis to *C. pomonella* egg laying [133]. Moreover, the spraying of glucose or fructose significantly reduced the percentage of damaged fruits by *C. pomonella* by 70% compared to the untreated control, with an effectiveness comparable with the spraying of the chemical insecticide deltamethrin [134,135].

In field trials, sucrose treatment was found to be as efficient as thiacloprid treatment in the reduction of damage by *C. pomonella*. Furthermore, synergistic effects were found when sucrose was combined with the thiacloprid insecticide [133], and between fructose and organophosphorus or insect growth regulator insecticides against the codling moth [136].

The quantities and ratios of soluble carbohydrates and on the leaf surface could also influence the egg-laying preferences of *Ostrinia nubilalis* on maize hybrids [137,138,139,140,141]. A study contributed to explore the efficacy of sucrose and fructose, used alone or in combination with natural pyrethrum, against *O. nubilalis* and *Scaphoideus titanus* [142]. The authors found that the application of sucrose associated with fructose provided the best efficacy in reducing the number of corn borer larvae per plant with a 23% efficacy. In the case of *S. titanus*, sucrose seemed to increase the action of natural pyrethrum, whilst the fructose showed the same efficacy as the natural pyrethrum.

Finally, sugars could be also interesting as components of commercial biopesticides due to the phagostimulant activity for a more effective ingestion by larvae [143]. These studies demonstrate a promising alternative to conventional crop protection tools [144] and pave the way for the development of eco-friendly control strategies using the new concept of “Sweet Immunity” induction.

### 3.3. Talc

Magnesium hydrogen metasilicate, known by the common name of talc, is approved as a basic substance to be used in outdoor applications on grapevines and fruit orchards to act as a physical barrier towards insects and mites, like *Cacopsylla pyri*, *Cacopsylla fulguralis*, *Drosophila suzukii*, *Panonychus ulmi*, and *Bactrocera oleae* [145,146,147].

Nowadays, the research interest in the use of inert dusts and their potential role in agriculture to manage diseases and protect crops from insect pests is increasing [148,149]. Among mineral products, natural zeolites, a broad range of crystalline hydrated aluminosilicates [150,151], could represent potential basic substances. Thanks to their physical and chemical properties and uses, the Codex Alimentarius Commission (1999) endorsed their use for pest control in food commodities and listed zeolites as granted substances in the organic food production and plant protection [152]. The insecticidal activity of zeolites towards stored-product insect pests, such as *Sitophilus zeamais*, *Rhyzopertha dominica*, *Sitophilus oryzae*, *Tribolium castaneum*, *Lasioderma serricorne*, *Tribolium confusum*, *Callosobruchus maculatus*, and *Meligethes* spp., was intensively reported [149,153,154,155,156,157,158,159,160,161,162]. In addition, the 40% reduction in the oviposition rates of *B. oleae* females due to zeolite applications was observed [163].

### 3.4. Diammonium Phosphate

Plant volatile compounds are involved in the host-finding process and oviposition site selection by insects [164,165]. The efficiency of traps used in indirect (e.g., monitoring) and direct (e.g., mass trapping, attract, and kill) semiochemical-based control tools can be improved significantly through the addition of certain food attractants [166]. Ammonia-releasing substances play an important role in both sexes of fruit fly attraction to food sources [167,168,169]. Thus, ammonia bait traps are currently used for monitoring fruit fly populations [170]. The use of diammonium phosphate is permitted to bait one trap per tree in orchards, including *Prunus* spp., *Citrus* spp., and olives (*Olea europaea*), to enable the massive capture of adults of the above-mentioned insect species. In this context, fruit fly pheromones added to food attractants, such as diammonium phosphate, are efficient for the monitoring and mass trapping of *C. capitata*, *B. cucurbitae*, and *B. dorsalis* [171], and are commonly used in the monitoring of *B. oleae* in most olive-growing countries of the Mediterranean basin [172].

### 3.5. Onion Oil

Onion oil obtained from *A. cepa* is authorized as a basic substance due to its repellent and scent masking activity against the carrot root fly, *Psila rosae* [173]. Dispensers of undiluted oil placed in the field are able to disorient adult flies which cannot find its host plant. Dispensers are filled with onion oil alone or with ethylene vinyl acetate granules that are able to improve the release of vapor.

### 3.6. Chitosan

Among these basic substances, chitosan stimulates the defense system of crops against several classes of pathogens, including fungi, viruses, bacteria, and phytoplasmas [22], and its use as an elicitor of the crop’s self-defense mechanisms has also been approved. Chitosan also exhibits a strong level of insecticidal activity against various insect pests [174]. The insecticidal activity of chitosan and its derivatives was demonstrated against the lepidopterans *Spodoptera littoralis* [78,175], *Helicoverpa armigera*, and *P. xylostella*, and the aphids *Aphis gossypii*, *Metopolophium dirhodum*, *H. pruni*, *Rhopalosiphum padi*, *Sitobium avenae*, and *M. persicae* [176,177]. The mortality of six types of aphids generally ranged between 60% and 80%, with a peak of 99.7%. Furthermore, recent studies showed that a new chitosan derivative, named avermectin-grafted-N,O-carboxymethyl chitosan (NOCC), showed an excellent insecticidal and acaricidal activity against *Aphis fabae*, *Nilaparvata lugens*, and *Tetranychus cinnabarinus* [178].

## 4. Basic Substances as Partners in Disease Management

The use of multiple protection products as alternatives or in combination with plant protection products is an increasing practice in agriculture. However, it is important to ensure that the combination of these substances does not lead to antagonistic effects that could reduce the efficacy of protection. It has often been suggested that an effective alternative to synthetic fungicides could be found through multiplying the protection product types [179], for example, the application of both a biocontrol microorganism and a plant extract showing antimicrobial activity. The question that this type of suggestion raises is compatibility [180], meaning, for example, that associating a substance with antimicrobial activity and a microorganism-based substance [181,182], or that mixing two biocontrol microorganisms [183] may lead to additive, synergistic, or even antagonistic effects.

Assessing a mixture or a treatment alternation efficiency is therefore important in order to ensure that satisfactory levels of protection are achieved [184]. Trials have been conducted to decipher which mechanisms of actions can be used together. They revealed that mixing microorganisms and resistance inducers increases the level of protection [185,186]. The application of basic substances in mixture with plant protection products can also lead to synergistic effects in pest management, as observed in mixing sucrose with pyrethrum in the control of *S. titanus,* the vector of the grapevine Flavescence dorée phytoplasma, which is an economically relevant pathogen [142]. Chitosan is an interesting basic substance that presents resistance-inducing activity [187,188] and enhances protection levels against pathogens on several crops when applied alongside different microorganisms, like *Trichoderma* spp. [189] or *Bacillus* spp. [190,191,192]. Chitosan possesses interesting features regarding their compatibility with biocontrol microorganisms, and also showed positive effects when used as a seed coating [193,194]. When used together, chitosan and microbial biocontrol agents can present an additive and, at times, have a synergistic effect, which means they could be interesting alternatives to synthetic fungicides for plant protection against fungal pathogens. When used in combination with copper, chitosan proved to be a useful tool for protecting grapevines from downy mildew, allowing for a reduction in copper doses and significant contributions to reducing copper inputs that are particularly important in organic farming.

Chitosan is, therefore, a promising basic substance that not only possesses interesting features regarding their compatibility with biocontrol microorganisms, but also shows positive effects when used as a seed coating. When chitosan and microbial biocontrol agents are used together, they exhibit a synergistic effect, making them a potential alternative to synthetic fungicides for protecting lettuce against fungal pathogens. Further research is, however, necessary to determine the optimal combination of protection products for different crops and pathogens/pests.

## 5. Potential Basic Substances: Approval Procedure and Issues

Along with the 24 approved basic substances, there are potential new solutions in this category, such as those listed in the new basic substance applications and extensions of these uses proposed for the existing basic substances. We maintain a list of ongoing basic substance applications for the Euphresco *Basics* program [16] and a list of ongoing basic substances applications [16,195]. The number of earlier deposits before April 2021 and the new procedure for basic substance application deposits and follow up [196] is still quite significant (Table 2). Since the new procedure was implemented, only a few basic substance applications and extensions have been submitted (Table 3), and some are currently under consideration. It appears that the International Uniform Chemical Information Database (IUCLID) procedure [197], which is more complicated and difficult to operate, makes applications more constraining for petitioners, at least in the filing and admissibility part. This situation, coupled with the recent non-approvals [16,195], favors using the usual structures for submitting the applications of active substances (consulting companies) and undoubtedly discourages new candidates, particularly those which are represented by smaller corporations, such as farmers, farmer associations, and small cooperatives, who used to apply in the past years.

Beyond the ongoing basic substances applications, we have identified several potential basic substances that have been considered or suggested by the EU member states, particularly via the coordination of minor uses. They include fennel (*Foeniculum vulgare*) oil, certain alcohols (such as 2-propanol), calcium chloride, or salicylic acid, although the latter is classified as a potential endocrine disruptor and has little chance of success.

In a second panel of candidates, certain previously approved active substances (excluding microorganisms) could be remobilized into basic substances, as has already been performed with pepper dust substances and sodium hypochlorite. Some of the candidates in this category include fenugreek (*Trigonella foenum-graecum*), ammonium acetate, seaweed extracts (except *Laminaria* spp.), citronella oil (*Cymbopogon* spp.), giant knotweed (*Reynoutria sacchalinensis*) extract, soybean (*Glycine max*) extract, wheat (*Triticum aestivum*) gluten, or gelatine, among others. Additionally, there are currently over 20 unapproved basic substances that are being considered for approval, which is a legal provision that has been specified in all the implementing regulations for the non-approval of each basic substance.

An attempt to resubmit an unapproved basic substance has already been unsuccessfully performed with pepper (*Capsicum annuum*) oleoresin and is currently under way with *Capsicum* oleoresin, proposed for a third attempt under the IUCLID, in view of the total orphan use that has taken place since the first attempt.

Apart from these substances which come easily to mind, since they have already been mentioned in one category or another of the current plant protection product Regulation EC 1107/2009 [9] or of its previous Directive EEC 91/414, much newer substances can be mentioned: some sugars (glucose, maltose, tagatose, etc.) have the crop protection properties described and could be the subject of further field efficacy trials and ultimately be submitted as a basic substance. Many plant extracts, essential oils, and even floral waters have been described in the literature. A few botanical substances have been suggested since 2014 when the first basic substance approvals began; in particular, plants of the Fabaceae family, which are well known for their antifungal properties, such as extracts of licorice (*Glycyrrhiza glabra*), alfalfa (*Medicago sativa*), or rumex (*Rumex crispus*), or other extracts, such as buckthorn (*Rhamnus* spp.), cinnamon (*Cinnamomum verum*), hop (*Humulus lupulus*), or ivy (*Hedera helix*).

It is also possible to consider some minerals, even though the commission’s stance on this is not entirely clear, especially regarding natural substances of mineral origin that act as physical barriers. The recent proposal of chabazite as a basic substance was voluntarily withdrawn after it was determined that it was not covered with the plant protection product regulations, even though kaolin, quartz sand, and especially talc, a basic substance authorized for organic farming [198], have been approved for this same plant protection product regulation [197]. Other minerals, such as acetate salts, could be effective against the black rot of grapevines (caused by the ascomycete *Phyllosticta ampelicida*), a resurgent disease since the reduction in the authorized quantities of copper in organic farming, the global decline in chemical pesticides [199], and the general reduction in treatments on vines resistant to fungal diseases (powdery and downy mildews).

Therefore, a number of potential basic substances could be used in crop protection due to their efficacy in controlling plant pathogens and pests. These include essential oils, alcohols, seaweed and plant extracts, sugars, minerals, and salts. In the future, it can be expected that new basic substances will be authorized. However, it must be pointed out that many potential basic substances (more than 20) have not been approved due to toxicological and ecotoxicological concerns, and this could discourage applicants. Another issue is the deterring effect of the quite complex IUCLID procedure, which is currently used for the submission and evaluation of applications for active substances. This may result in fewer smaller entities becoming applicants, in particular, while larger consulting firms who have more experience with the process could be better equipped to handle the complexity of the application process. If possible, the procedure to promote applications should be simplified. Furthermore, farmers may be unaware of the potential benefits of basic substance use, due to the limited advertising and technical information for field application. Information on basic substances employment is particularly useful for farmers from neighboring EU countries (for example Eastern Europe, North Africa, and the Middle East), which often trade their products in European countries. In this context, education, training, and technical advice to farmers regarding basic substances in sustainable pest management strategies are seriously needed [16].

## 6. Conclusions

The use of basic substances in crop protection is an area of active research, with several promising substances showing efficacy in controlling plant pathogens and pests. Therefore, it is expected that the number of basic substances available will increase in the future, even though the regulatory process for approving new basic substances can be complex and may discourage some applicants. Chitosan is currently the most commonly used and well-studied basic substance. Other substances, such as essential oils, alcohols, seaweed and plant extracts, sugars, minerals, and salts also show potential, alone or in combination/alternation with other plant protection products. However, they are not used very often in practice, since their field efficacy depends on various factors, including the crop, the pathogen or pest, the origin of the basic substance (e.g., type of extract, purity, and origin of the active substance), and the timing and method of application. The composition of the basic substance is highly relevant, as demonstrated through the different levels of protection achieved by red and white onion extracts on potato early blight. The combination of basic substances with biocontrol microorganisms and plant extracts with antimicrobial activity can be an effective alternative to plant protection products. However, concerns regarding the compatibility between these substances and potential antagonistic effects have been raised. Therefore, it is important to assess the effectiveness of such mixtures or treatment alternations to ensure satisfactory protection levels. To date, there is not enough knowledge on the use of approved basic substances to control insect pests, and further studies are needed to better understand their modes of action and to improve their application methods and timing. This is particularly important since basic substances could also represent effective tools in insecticide resistance management strategies due to their different modes of action, which are often associated with physical barriers or repellents and lure effects without biocidal activity [146].

Globally, the employment of basic substances and potential basic substances may be an encouraging support to control diseases and a help in satisfying the European Green Deal aims and reducing the application of synthetic plant protection products. However, due to the very specific efficacies found in different crop pests and diseases, field trials in specific environments and developing accessible technical advice to farmers regarding basic substances are key aspects for contributing to sustainable pest management strategies. 

In conclusion, further field experimentation is needed to fully understand the potential of basic substances as alternatives to, or partners of, plant protection products in different contexts. Field trials also lead to the validation of the most effective application strategies based on the operational conditions, focusing in particular on insect control, since this area is less explored. Nonetheless, these reviewed studies provide promising results on the use of basic substances in integrated pest management or organic farming strategies for reducing the environmental impact of crop protection.

## Figures and Tables

**Figure 1 plants-12-03152-f001:**
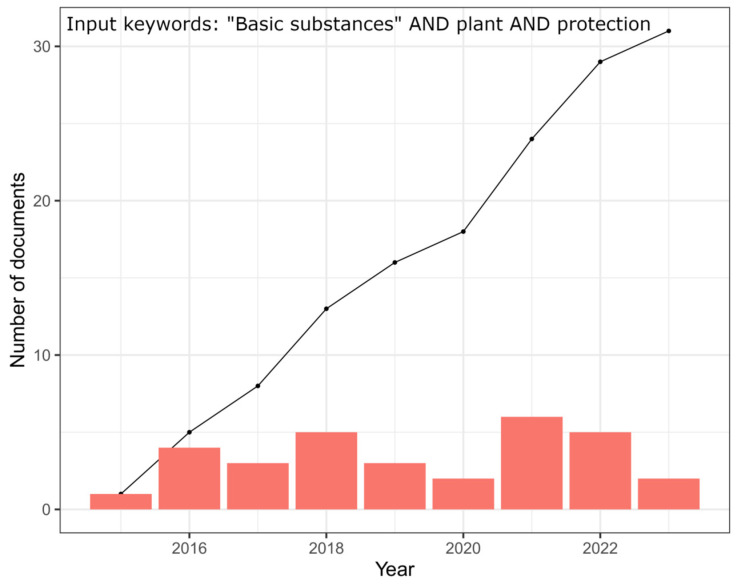
Number of scientific publications per year involving basic substances use in plant protection and total accumulated publications (line) over the 2015–2023 period. Source: Scopus database (accessed 28 March 2023).

**Figure 2 plants-12-03152-f002:**
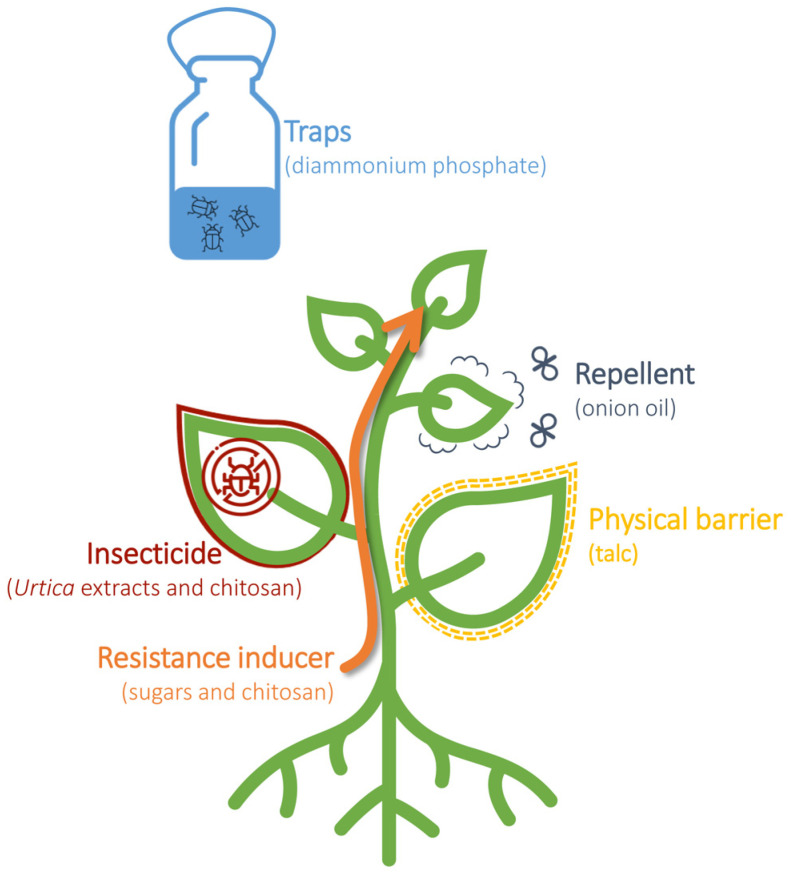
Utilization of basic substances in the management of plant pests. Basic substances can be used as insecticides, resistance inducers, physical barriers, repellents, and traps to control and manage plant pests.

**Table 2 plants-12-03152-t002:** Ongoing basic substance applications (BSAs) and extensions (Ext.) for approved basic substances.

Basic Substance	BSA or Extension	Regulatory Stage	Issue
NaCl	Ext.	Vote	Positive
Willow bark and stem extract	BSA	Vote	Negative
H_2_O_2_ silver stabilized	BSA	Vote	Negative
*Yucca schidigera* extract	BSA	Vote	Negative
CaOH_2_	Ext.	Vote	Stop clock
Sodium hypochlorite	BSA	Vote	Uncertain
Caffeine	BSA	Vote	Stop clock
Ozone	BSA	Vote	Stop clock
Chitosan HCl	Ext.	Vote	Uncertain
Sainfoin (*Onobrychis viciifolia* var. Perly) dried pellet	BSA	EFSA outcome	To be determined
*Quassia amara*	BSA	EFSA outcome	To be determined
Magnesium hydroxide	BSA	EFSA outcome	To be determined
*Moringa oleifera*	BSA	Submitted	Questions for admissibility
*Psidium guajava* L. leaf extract	BSA	Submitted	Questions for admissibility
Organic polyphenolic botanical compost	BSA	Submitted	Questions for admissibility
Grape seed extract	BSA	Evaluation	To be determined
*Allium fistulosum* extract	BSA	Evaluation	To be determined
Eggshell	BSA	Evaluation	To be determined
Water	BSA	Submitted	-
Pepper dust	BSA	Submitted	Questions for admissibility
*Ocimum gratissimum* extract	BSA	Submitted	Questions for admissibility
Chitosan	Ext.	Vote	Uncertain
*Equisetum arvense*	Ext.	Vote	Negative
NaCl	Ext.	Vote	Uncertain
*Urtica* sp.	Ext.	Submitted	Uncertain
*Urtica* sp.	Ext.	Submitted	Uncertain
Sunflower oil	Ext.	Submitted	Abandoned
*Equisetum arvense*	Ext.	Submitted	Abandoned
Lecithin	Ext.	Submitted	Abandoned
*Salix* cortex	Ext.	Submitted	Uncertain

**Table 3 plants-12-03152-t003:** Ongoing basic substance applications (BSAs) and extensions (Ext.) under the IUCLID procedure.

Basic Substance	BSA or Extension	Regulatory Stage	Issue
Ginger extract	BSA	Submitted	Questions for admissibility
*Capsicum* oleoresin	BSA	Submitted	Questions for admissibility
Vinegar	Ext.	Submitted	
*Plectranthus amboinicus*	BSA	Submitted	
*Plantago major*	BSA	Submitted	
NaCl	Ext.	Ongoing	

## Data Availability

No new data were created or analyzed in this study. Data sharing is not applicable to this article.

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
