# Peer review of "Are Basic Substances a Key to Sustainable Pest and Disease Management in Agriculture? An Open Field Perspective"

_plants, 2023, doi:10.3390/plants12173152_

Round 1

Reviewer 1 Report

The latest related literatures should be added into the text. 

The text needs to polished, due to some sentences were difficult to understand.

Author Response

Dear reviewer, thanks for the comments.

We added the latest information available on the subject. Can you please specify to which parts of the manuscript you’re referring to?

The manuscript has been revised by an English mother tongue before submission. Can you please be more specific on the phrases that were difficult to understand? In this way we could try rephrasing them.

Sincerely,

Silvia Toffolatti

Reviewer 2 Report

Dear authors,

I find your review paper on basic substances very interesting and relevant. It was very useful reading demonstrating that basic substances can be as efficient as classical chemical pesticides under many conditions. Having said that, I think your manuscript would benefit of some revisions before it can be published. This has mainly to do with the structure of the paper. The way it is structured now, you get quite a lot of repetition of the same statements throughout the document. This is especially true in chapters 2.1.1 and 2.1.2 about chitosan. E.g. the mode of action is repeated and also what factors that can affect chitosan’s efficiency. For reference, please look at lines 151-158 and 214-226 and you might see what I mean. Another example is lines 163-166 and 242-245, where also the same reference (42) is made. In chapter 2.3 on preharvest treatment some of the same information about chitosan on grapes appears again, illustrating the same issue. In addition, I want to raise the question if chapters 2.4 is necessary at all? Much repetition, and of which some could be moved to a general summary at the end of the paper. I think much could be benefitted from merging those two chapters together by putting the same type of information together and removing repetitions. You see the same tendencies of repetition and information that could be put together in one place, throughout the manuscript. This may be the result of different people writing the different parts without necessarily looking too close on what has been written by others. The message is to read the manuscript thoroughly and wash out repetitions and put information that belong together in the same paragraph.

Another general comment is the inconsistence in the use of common names and latin names. Sometimes only the common names are used, at other times the latin name, and sometimes both. Please be consistent in the way you use the names of the different species. One suggestion is to use common names, but then include the latin name the first time you mention the species. Or include both always.

Other more specific comments:

Line 240: “chitosan-treated "Chardonnay" wine grapes exhibited a degree of severity at harvest that was more than halved”. Severity of what? Should be “severity of damage” or something similar to be more precise.

Line 294: please include latin name of horsetail.

Lines 371-372: Written references and not numbered. And question mark behind one of them. I suggest you check your reference list one more time as well.

Chapter 3: A lot of results on efficiency are mentioned, but no numbers as in chapter 2. Numbers may help to illustrate how efficient the substances are against the different pests.

Line 457: Explain MOA abbreviation.

Line 511: proline as in Proline the plant protection product? Please clarify this in the text.

Lines 513 and 538: References are written in the form xxx and coworkers/collaborators. In my opinion this should be avoided in the case where you refer by using numbers. It would be more suitable if you write the references in the xxx et al. (2023) - format throughout the manuscript.

Chapters 3.7 and 4: Necessary? Some repetition and maybe some of it can be moved elsewhere or to the general summary.

Line 657: “members”. Do you mean Member States of the EU? Please clarify.

Author Response

Dear reviewer,

we are grateful for your comments.

Please find below and in attachment the response to your comments. 

We think that themanuscript greatly improved thanks to your suggestions.

Sincerely,

Silvia Toffolatti

Reviewer 2 (R2).

Comments and Suggestions for Authors

Dear authors, I find your review paper on basic substances very interesting and relevant. It was very useful reading demonstrating that basic substances can be as efficient as classical chemical pesticides under many conditions. Having said that, I think your manuscript would benefit of some revisions before it can be published.

Authors. Thanks for the comments, we appreciate your help and think that the revised version of the manuscript greatly improved thanks to your suggestions.

R2. This has mainly to do with the structure of the paper. The way it is structured now, you get quite a lot of repetition of the same statements throughout the document. This is especially true in chapters 2.1.1 and 2.1.2 about chitosan. E.g. the mode of action is repeated and also what factors that can affect chitosan’s efficiency. For reference, please look at lines 151-158 and 214-226 and you might see what I mean. Another example is lines 163-166 and 242-245, where also the same reference (42) is made. In chapter 2.3 on preharvest treatment some of the same information about chitosan on grapes appears again, illustrating the same issue.

Authors. The following parts were eliminated from Par. 2.1.2: “Chitosan, a linear polymer of D-glucosamine that can be derived from crab shells, fun-gi and insects [61,62], is available on the market in different formulations. Chitosan properties reflect the composition, which depends not only on the origin but also on the deacetylation degree, molecular weight, production process (biological or chemical synthesis), and environmental parameters such as the pH and temperature [60,63,64]. Its water solubility, bio-adhesive properties, and user-friendliness, like its antifungal efficacy, vary according to the above-mentioned properties and hydrochloride degree. Three different pathways are involved in chitosan effectiveness: i) elicitation of host defence mechanisms, ii) direct antimicrobial activity, and iii) coating properties. Chi-tosan hydrochloride also has direct and indirect antifungal effects [60,64].”; “In general, no phytotoxic effect of chitosan has been reported on products, except for a slight shining of treated tissues [71], whereas some toxicity was observed on leaves using high concentrations of chitosan, as observed on grapes [43].” In par. 2.3, the following phrase “In case of table grapes (V. vinifera), grey mould and other decays were decreased dur-ing storage, following 0.1-1 % chitosan application before harvest alone, or in combination with UV-C treatment. Similarly, the preharvest application of 0.2-1 % chitosan was effective against grey mould latent infection and decay of strawberries (Fragaria x ananassa and Fragaria chiloensis) was changed into “Similarly to what observed on table grape [97–99], the preharvest application of 0.2-1 % chitosan was effective against grey mould latent infection and decay of strawberries (Fragaria x ananassa and Fragaria chiloensis)”.

R2. In addition, I want to raise the question if chapters 2.4 is necessary at all? Much repetition, and of which some could be moved to a general summary at the end of the paper. I think much could be benefitted from merging those two chapters together by putting the same type of information together and removing repetitions. You see the same tendencies of repetition and information that could be put together in one place, throughout the manuscript. This may be the result of different people writing the different parts without necessarily looking too close on what has been written by others. The message is to read the manuscript thoroughly and wash out repetitions and put information that belong together in the same paragraph.

Authors. Par. 2.4 was deleted. Only a few parts at the end of the paragraph were kept and moved to the Conclusions. Please, see the manuscript for modifications.

R2. Another general comment is the inconsistence in the use of common names and latin names. Sometimes only the common names are used, at other times the latin name, and sometimes both. Please be consistent in the way you use the names of the different species. One suggestion is to use common names, but then include the latin name the first time you mention the species. Or include both always.

Authors. As suggested, we placed the scientific name and common name the first time a plant species was cited. Then we used the common name. Examples: “grapevine (Vitis vinifera)” in par. 2; “nettle (Urtica spp.)” in par. 2.2. For pathogens and insects, only scientific names were used.

Other more specific comments:

R2. Line 240: “chitosan-treated "Chardonnay" wine grapes exhibited a degree of severity at harvest that was more than halved”. Severity of what? Should be “severity of damage” or something similar to be more precise.

Authors. Thanks for signalling this problem. We were referring to disease severity. “Severity” was changed into “disease severity”.

R2. Line 294: please include latin name of horsetail.

Authors. The latin name of horsetail was presented at the first time it was cited, at line 168: “Equisetum arvense (horsetail)”.

R2. Lines 371-372: Written references and not numbered. And question mark behind one of them. I suggest you check your reference list one more time as well.

Authors. The written references were converted to numbers (93 and 94).

R2. Chapter 3: A lot of results on efficiency are mentioned, but no numbers as in chapter 2. Numbers may help to illustrate how efficient the substances are against the different pests.

Authors: The efficiency levels were added in the text. Please see lines 676, 683, 758, 809-810.

R2. Line 457: Explain MOA abbreviation.

Authors. “MOA” was changed to “mode of action”.

R2. Line 511: proline as in Proline the plant protection product? Please clarify this in the text.

Authors. The authors of the cited work found that the amino acid proline had a synergistic effect with sugars. “proline” was changed into “the amino acid proline”.

R2. Lines 513 and 538: References are written in the form xxx and coworkers/collaborators. In my opinion this should be avoided in the case where you refer by using numbers.

Authors. The phrases were changed to avoid “coworkers/collaborators).

R2. It would be more suitable if you write the references in the xxx et al. (2023) - format throughout the manuscript.

Authors. We used the journal format for citations. Mendeley software was used to manage the citations.

R2. Chapters 3.7 and 4: Necessary? Some repetition and maybe some of it can be moved elsewhere or to the general summary.

Authors. Chapter 3.7 was deleted. Only a part of it (for instance, this part: ”to date, there is not enough knowledge on the use of approved basic substances to control insect pests, and further studies are needed to better understand their modes of action and to improve application methods and timing. This is particularly important since basic substances could also represent effective tools in insecticide resistance management strategies due to their different modes of action, often associated with physical barrier or repellent and lure effects without biocidal activity [147,180].”) was placed in the Conclusions. We prefer to keep chapter 4 separated from the others, because it explains important concepts that should be summarized too much in the context of the conclusions.

R2. Line 657: “members”. Do you mean Member States of the EU? Please clarify.

Authors. Yes: we changed “members” with “EU Member States”.

Reviewer 3 Report

I would like to express my gratitude to the authors for this review. It is very interesting, including many contemporary most recent literary references, as well as references to time-tested classic highly cited papers. This balance directly reflects the good scientific taste, the high general scientific level of the authors, as well as their skill. I received great intellectual pleasure from reading this review, which is written logically, consistently, and methodologically correct. The review also contains interesting information, a qualitative and comprehensive analysis of this information. In current reviews, one can rarely find the author's attitude to the problems described, this happens only in very good reviews made by real professionals. This is just such a review.

There is a lot to be said for this review. But it has only one major drawback. This is the number of authors. 17 authors!!! How is that? It turns out that each author wrote only less than 1 page of text. The authors are smart people and I'm sure they understand that this fact looks ridiculous. I propose that this review be accepted for publication in its present form. And I recommend to the authors to reduce the creation team at their discretion, but I do not protest if they leave the entire composition of the authors. I just ask you to take my opinion into account at least in the future. If no one tells you about it, it does not mean that no one notices it.

Author Response

Dear reviewer, 

we thank you for the precious comments.

It is true that many authors are involved in the manuscript, but this is due to the fact that the review initiative was born within the context of the Euphresco project “Basics” which deals with the efficacy of basic substances for the control of diseases and pests at the field level". I'm the leader of WP3, which is dealing with the assessment of efficacy of basic substances at the field level, and I also coordinated the writing of the manuscript. As you can see below in the author contribution statement, each partner had a role in the manuscript writing. As a consequence, we are not able to reduce the number of authors.

The author contributions, where the work done by each author is specified, and the funding information, where it is specified that the authors participated to the Basics project, were already present in the manuscript. 

Author Contributions: S.L.T.: conceptualization, writing - original draft, writing - review and ed-iting, and supervision. Y.D.: writing - original draft (paragraph 4). I.D.I, C.F.,G.G., N.M: writing – original draft (paragraph 3). A.I., Y.K., A.M.: writing – original draft (paragraph 2.1.2). J.K.: writing – original draft (paragraph 2.2). G.M.: writing - review and editing, and visualization. P.M.: writing – original draft (paragraph 5). D.M.: writing – original draft (paragraph 1), and visualization. K.M.,E.S.: writing – original draft (paragraph 2.3). S.P.: writing – original draft (paragraph 2.1.1). G.R.: conceptualization, writing - original draft (paragraph 2.1.1), writing - review and editing.

Funding: The authors of this work participated to the collaborative Euphresco project “Basics - Basic substances as an environmentally friendly alternative to synthetic pesticides for plant protection”.

Following your suggestion, we specified that the review was born within the Basics projectin the acknowledgements section of the manuscript. Please, find below the details.

Acknowledgments: This work originated within the context of WP3 “Testing basic substances and potential basic substances to manage diseases and pests in the field” in the Euphresco project “Basics - Basic substances as an environmentally friendly alternative to synthetic pesticides for plant pro-tection” The authors wish to thank the Euphresco Coordinator at EPPO, Baldissera Giovani, for the support in the project management.

Yours sincerely,

Silvia Toffolatti